

# Comparison of OMI $NO_2$ observations and their seasonal and weekly cycles with ground-based measurements in Helsinki

Iolanda Ialongo[1], Jay Herman[2], Nick Krotkov[2], Lok Lamsal[2,3], Folkert Boersma[4,5], Jari Hovila[1], and Johanna Tamminen[1]

[1]Earth Observation Unit, Finnish Meteorological Institute, Helsinki, Finland
[2]Atmospheric Chemistry and Dynamics Laboratory, NASA Goddard Space Flight Center, Greenbelt, Maryland, USA
[3]GESTAR, Universities Space Research Association, Columbia, Maryland, USA
[4]KNMI - Climate Observations Department, De Bilt, Netherlands
[5]Wageningen University - Meteorology and Air Quality Group, Netherlands

*Correspondence to:* I. Ialongo (iolanda.ialongo@fmi.fi)

**Abstract.** We present the comparison of satellite-based OMI (Ozone Monitoring Instrument) $NO_2$ products with ground-based observations in Helsinki. OMI $NO_2$ total columns, available from standard product (SP) and DOMINO algorithm, are compared with the measurements performed by the Pandora spectrometer in Helsinki in 2012. The relative difference between Pandora #21 and OMI SP retrievals is 4% and -6% for clear sky and all sky conditions, respectively. DOMINO $NO_2$ retrievals showed slightly lower total columns with median differences about -5% and -14% for clear sky and all sky conditions, respectively. Large differences often correspond to cloudy autumn-winter days with solar zenith angles above 65°. Nevertheless, the differences remain within the retrieval uncertainties. Furthermore, the weekly and seasonal cycles from OMI, Pandora and $NO_2$ surface concentrations are compared. Both satellite- and ground-based data show a similar weekly cycle, with lower $NO_2$ levels during the weekend compared to the weekdays as result of reduced emissions from traffic and industrial activities. Also the seasonal cycle shows a similar behaviour, even though the results are affected by the fact that most of the data are available during spring-summer because of cloud cover in other seasons.

This is one of few works in which OMI $NO_2$ retrievals are evaluated in a urban site at high latitudes (60°N). Despite the city of Helsinki having relatively small pollution sources, OMI retrievals have proved to be able to describe air quality features and variability similar to surface observations. This adds confidence in using satellite observations for air quality monitoring also at high latitudes.

## 1 Introduction

Nitrogen oxides ($NO_x$ = NO + $NO_2$) play an important role in tropospheric chemistry, participating in ozone and aerosol production processes. $NO_x$ is mainly generated in polluted regions by anthropogenic combustion and it is toxic when present at high concentrations at the surface.

The $NO_2$ content in atmosphere can be monitored using satellite observations. Satellite-based $NO_2$ total and tropospheric columns have been available since 2004 from the Dutch–Finnish Ozone Monitoring Instrument (OMI), onboard NASA's EOS


(Earth Observing System)-Aura satellite (Levelt et al., 2006). OMI provides almost-daily global coverage with nominal spatial resolution of $13 \times 24\,\mathrm{km}^2$ at nadir.

Satellite instruments provide global $NO_2$ observations used in several air quality applications including recent studies on emission and lifetime estimation (Beirle et al., 2011; de Foy et al., 2015; Lu et al., 2015; Liu et al., 2016; McLinden et al.,
2016), emission changes (Castellanos and Boersma, 2012; McLinden et al., 2012; Hilboll et al., 2013; Duncan et al., 2013; Krotkov et al., 2016), ship emission monitoring (deRuyter de Wildt et al., 2012; Ialongo et al., 2014), satellite constrained $NO_x$ emission inventory (Lamsal et al., 2011; Ghude et al., 2013; Streets et al., 2013; Vinken et al., 2014). Also, satellite data have been used for urban pollution monitoring, e.g. looking at the $NO_2$ weekly cycle (Beirle et al., 2003; Boersma et al., 2009). The results of these studies are strongly affected by the accuracy of the satellite retrievals thus, accurate validation against
independent ground-based measurements is continuously needed.

Recently, the Pandora instrument has been developed to help in evaluating satellite $NO_2$ retrievals with ground-based measurements (Herman et al., 2009). The Pandora spectrometer system measures direct sunlight in the UV-VIS spectral range (280–525 nm). It provides $NO_2$, $O_3$ and $SO_2$ total columns through direct-sun DOAS (Differential Optical Absorption Spectroscopy) technique. This technique provides very accurate $NO_2$ observations, compared to zenith sky measurements, because
it does not require complex prior assumptions for converting the slant to the vertical columns. Because Pandora is a low cost instrument, it is largely applied for satellite-data validation and the observation network is quickly growing (see http://acdb-ext.gsfc.nasa.gov/Projects/Pandora/index.html).

Lamsal et al. (2014) extensively evaluated the current version of OMI $NO_2$ retrievals using several different ground-based observations, including Pandora measurements. They found that OMI and Pandora $NO_2$ total columns are fairly correlated
(r=0.25) and in agreement to within 30%. Before that, Pandora measurements have been used for evaluating OMI $NO_2$ total columns also by Herman et al. (2009) and Tzortziou et al. (2013). Also, Knepp et al. (2013) estimated surface $NO_2$ mixing ratios from Pandora measurements and found high correlation (typically r>0.75) with surface records from a photolytic-converter-based instrument.

Most of the validation studies are performed at middle-low latitude sites and a detailed evaluation of OMI $NO_2$ products at
higher latitudes is still missing. High latitudes, and in particular the Arctic, are becoming more and more important because of the increasing anthropogenic activities foreseen in these regions (e.g., new oil extraction and mining sites, new shipping routes as well as urban emissions). Satellite-based observations offer unique opportunity for monitoring atmospheric composition in such remote areas with very sensitive environment. Thus, the quality of atmospheric observations needs to be continuously evaluated in order to provide reliable retrievals.
This work aims at evaluating the quality of OMI $NO_2$ products through comparison with ground-based observations in Helsinki (Finland), which is the northernmost city (latitude of 60.2°N) with more than half a million inhabitants. The database used in the analysis is described in Sect. 2. The results of the comparison of OMI $NO_2$ total columns with ground-based Pandora observations are shown in Sect. 3.1. Also, OMI $NO_2$ seasonal and weekly cycle are compared to those derived from surface concentrations from air quality station in Sect. 3.2. Finally, the summary and conclusions are presented in Sect. 4.





## 2 NO$_2$ observations

### 2.1 OMI NO$_2$ products

In this work, OMI NO$_2$ total and tropospheric column densities are taken into account. OMI is a Dutch-Finnish instrument operating on-board NASA's Aura satellite since October 2004. OMI measures solar backscattered light in the UV-VIS spectral region using a 2-dimentional CCD detector. The cross-track swath is divided into 60 pixels. The nominal resolution at nadir (row 30) is $13{\times}24$ km$^2$, with increasing pixel size towards the edges of the swath (up to $28{\times}150$ km$^2$). The Aura satellite flies in a Sun-syncronous polar orbit with nominal equator crossing time 13:45 LT with almost daily global coverage. At high latitudes more than one daily overpass can be obtained because of the overlapping orbits. Since 2007 the so-called "row anomaly" affected some of the cross-track positions of the swath, reducing the spatial coverage of the instrument. In this work, the affected rows are removed according to the operational flagging for the row anomaly.

OMI NO$_2$ retrievals are obtained from the spectral measurements in the visible between 405 and 465 nm. There are two NO$_2$ products available from OMI: NASA's standard product (SP) version 2.1 (Bucsela et al., 2013) and KNMI's (Royal Netherlands Meteorological Institute) DOMINO (Derivation of OMI tropospheric NO$_2$) product version 2 (Boersma et al., 2011). Both retrievals are based on DOAS technique but they differ on the way of converting slant columns into the vertical columns. Moreover, the separation between stratospheric and tropospheric columns is different. In SP algorithm the stratosphere-troposphere separation is based on the OMI observations over areas with relatively little tropospheric NO$_2$, while DOMINO algorithm assimilates OMI observations into chemistry-transport model. Comprehensive validation of the latest version of OMI retrievals with independent measurements was presented by Lamsal et al. (2014) (and references therein). They showed that OMI retrievals are lower in urban regions and higher in remote areas, but generally in agreement with ground-based and airborne measurements within $\pm 20\,\%$.

The Helsinki overpass SP data (available at http://avdc.gsfc.nasa.gov/) are taken into account in this study. The NO$_2$ retrievals from DOMINO product corresponding to the same pixels as in the SP overpass file are obtained from TEMIS website (http://temis.nl). The retrieval uncertainty for OMI NO$_2$ vertical column is in the order of $10^{15}$ molec. cm$^{-2}$ for the Helsinki overpass dataset. Cloud fraction (CF) data from OMI are used to identify the almost cloud-free scenes. Both OMI NO$_2$ retrieval algorithms include as input information the OMCLDO2 cloud product, which is based on the O2-O2 absorption method (Acarreta et al., 2004).

The mean NO$_2$ tropospheric column in Helsinki during May-September 2005–2014 are shown in Fig. 1. The mean NO$_2$ tropospheric column value goes up to about $3{\times}10^{15}$ molec. cm$^{-2}$. This is about five times smaller than what can be observed for example in central Europe and it is close to the OMI detection limit ($\pm 5{\times}10^{14}$ molec. cm$^{-2}$). In addition to Helsinki, the main polluting sources in this area are the cities of Tallin and Turku as well as emissions from ships in the Gulf of Finland.

### 2.2 Ground-based observations

OMI NO$_2$ total columns are compared against ground-based observations performed during 2012 in Helsinki-Kumpula station (60.20° N, 24.96° E), Finland by Pandora instrument #21. The measuring site is located approximately under the black dot





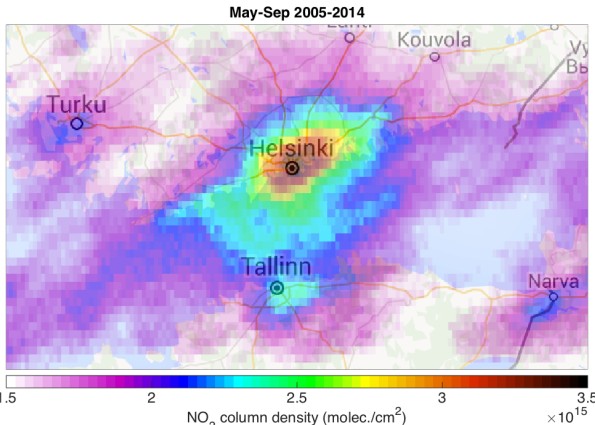

**Figure 1.** OMI $NO_2$ tropospheric column in Helsinki. The map shows the average over the time period 2005–2014 from May to September with $0.05° \times 0.05°$ spatial resolution.

corresponding to Helsinki in Fig. 1. The Pandora system includes a spectrometer connected by a fiber optic cable to a sensor head with $1.6°$ FOV (field of view). A sun-tracking device allows the optical head to point at the centre of the Sun with of $0.013°$ resolution. Pandora performs direct-sun measurements in the UV-VIS spectral range (280–525 nm) and provides $NO_2$, $O_3$ and $SO_2$ vertical column densities. The algorithm first derives the relative $NO_2$ slant columns by least-square fitting and

5   then converts to absolute values through the reference spectrum obtained by Langley-extrapolation technique. Pandora retrieval employs a temperature correction to the cross-sections used in the spectral fitting procedure (as also in OMI $NO_2$ retrieval). The $NO_2$ columns are available every about 1.5 minutes. The full description of the Pandora instrument and the algorithm for the inversion methodology is reported by Herman et al. (2009). The nominal clear-sky precision in the Pandora $NO_2$ total column retrieval is in the order of $3 \times 10^{14}$ molec. $cm^{-2}$ with an accuracy of about $\pm 1.3 \times 10^{15}$ molec. $cm^{-2}$. Ground-based

10   cloud cover information from celiometer located in Kumpula site (available at hav.fmi.fi) are used together with OMI CF cloud information, in order to identify the cloud-free scenes.

The $NO_2$ surface concentrations available in Helsinki-Kumpula air quality station were used for the analysis of the seasonal and weekly cycle. The surface concentration data are obtained from SMEAR database (Junninen et al., 2009) available online at: avaa.tdata.fi/web/smart/smear. Kumpula station is classified as semi-urban because it is influenced by car pollution only

15   downwind from the large traffic street. The surface $NO_2$ concentrations are measured using online trace level gas analyser based on the ultraviolet fluorescence method (i.e., European reference 5 method). Hourly average concentrations are used in this study. Only the measurements closest to the satellite overpass time (within 30 minutes) are taken into account. Note that Pandora spectrometer is located on the roof of FMI building, about 25 m above the air quality station (altitude about 4 m agl).





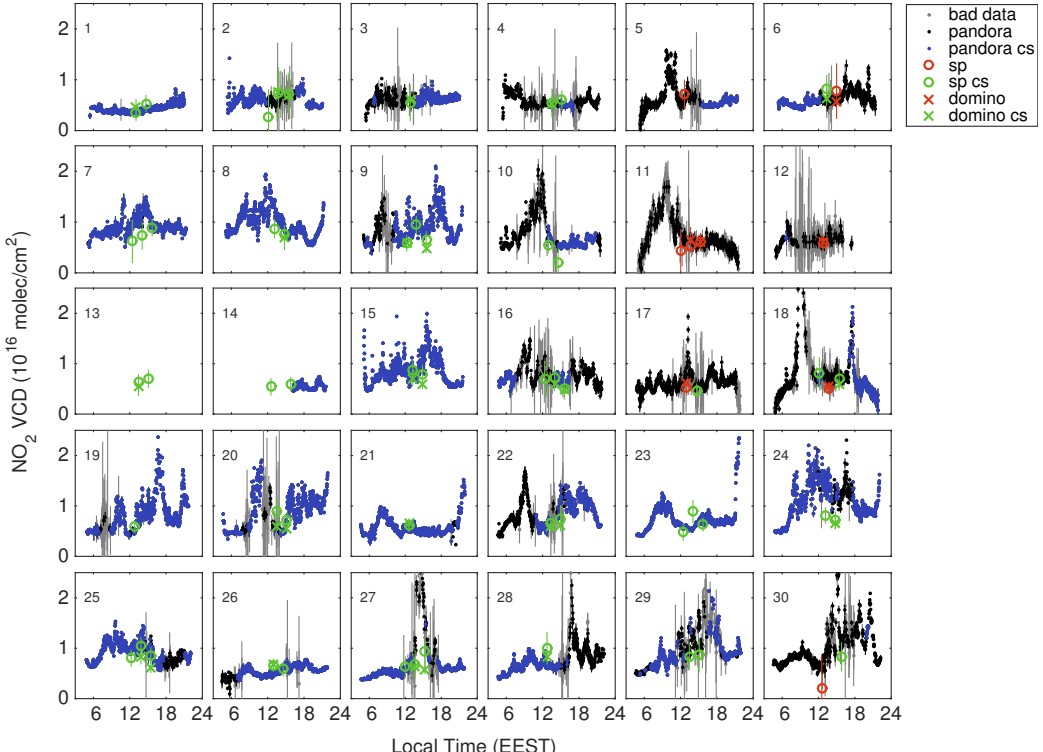

**Figure 2.** OMI and Pandora NO$_2$ total columns during May 2012. Both OMI SP and DOMINO data are shown. The day of the month is reported on the upper left corner of each subplot. OMI data are screened for clear sky conditions using OMI CF $<$ 0.5 (green circles and crosses for OMI SP and DOMINO, respectively), while Pandora clear-sky data (blue dots) are derived using cloud cover information from celiometer (below 5/8). Gray dots with vertical uncertainty bars indicate Pandora retrievals with uncertainties larger than $1.3{\times}10^{15}$ molec. cm$^{-2}$.

## 3  Results

### 3.1  Comparison of OMI NO$_2$ total columns with Pandora observations

Figure 2 shows an example of the NO$_2$ total columns from Pandora and OMI overpasses during May 2012. Both OMI SP and DOMINO retrievals are included, with the former usually showing larger values than the latter. Pandora retrievals with uncertainty larger than $1.3{\times}10^{15}$ molec. cm$^{-2}$ (gray dots in Fig. 2) are removed from the comparison shown in the following sections. OMI data are cloud-screened according to OMI CF (below 0.5) while Pandora measurements according to the ground-based cloud cover information from celiometer (below 5/8). These threshold values include clear-sky and partially cloudy scenes. These two cloud-screening criteria give similar results (see green symbols and blue dots in Fig. 2 for OMI and Pandora, respectively). When considering all the collocated data available in 2012, the cloud screening criteria agree in more than 80% of the cases.





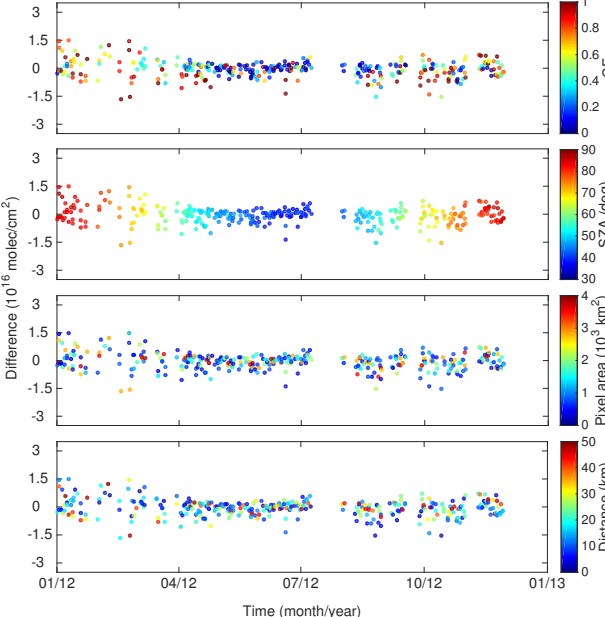

**Figure 3.** Difference between OMI SP and Pandora $NO_2$ total column in Helsinki during 2012. The colorscales in the different panels correspond to OMI CF, SZA, pixel area and distance between the actual location of Pandora instrument and the center of OMI pixel.

Figure 3 shows the difference between OMI SP and Pandora $NO_2$ total columns during 2012 as a function of CF, solar zenith angle (SZA), pixel area and distance between the city center and the center of the pixel. The median relative difference is (4±19)% and (-6±25)% for clear-sky and all-sky conditions, respectively. These percentage values correspond to absolute differences (3±11)×$10^{14}$ molec. cm$^{-2}$ and (-4±18)×$10^{14}$ molec. cm$^{-2}$, respectively. For the calculation of the clear-sky median both criteria based on OMI CF and ground-based cloud cover are used to screen for the cloudy scenes. A similar comparison for DOMINO product (Fig. S1 in the supplementary material) shows that the median relative difference is (-5±13)% and (-14±18)% for clear-sky and all-sky conditions, respectively (or in terms of absolute values (-3±9)×$10^{14}$ molec. cm$^{-2}$ and (-9±16)×$10^{14}$ molec. cm$^{-2}$, respectively). The semi-interquartile is used to calculate the variability of the difference. Part of the discrepancy is explained by the difference between the OMI pixel and the relatively smaller Pandora FOV. This effect is especially important when the ground-based station is downwind from a high traffic street.

It must be noted that there is a larger amount of valid retrievals available from SP product than from DOMINO (especially during winter). This is caused by the fact that DOMINO retrievals are not available for SZA larger than 80°. The different sampling only partly explain the observed difference between the median relative difference obtained from the two different OMI products. The remaining differences in the total columns from SP and DOMINO can be attributed to differences in air mass factor values (about 13% smaller for OMI SP) used to convert the slant to vertical columns. Because the slant columns from SP and DOMINO are very similar to each others, the total column values from DOMINO algorithm are also found to be about 13% smaller.




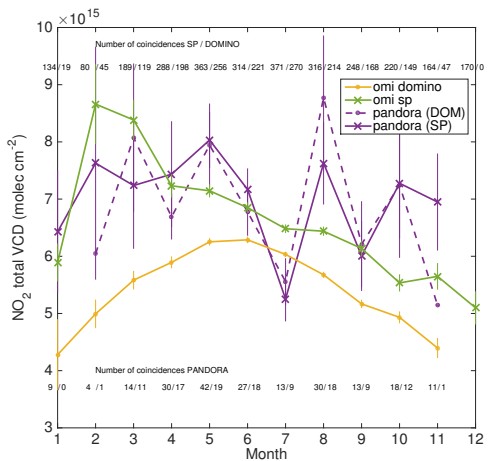
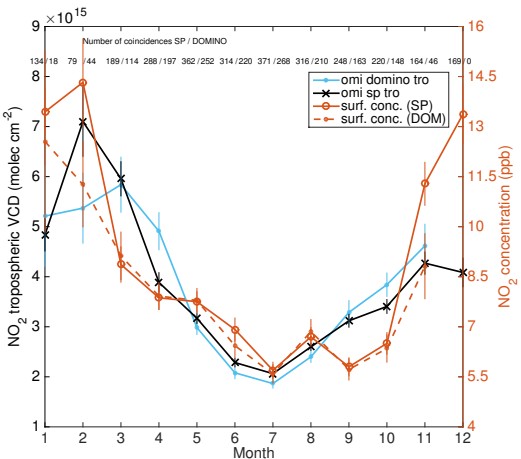

**Figure 4.** Left: NO₂ seasonal cycle from total columns from OMI SP (green) and DOMINO (yellow) products during the period 2006–2014. The monthly means from collocated Pandora NO₂ total columns measured in Helsinki during 2012 are also shown (purple). Right: NO₂ seasonal cycle from tropospheric columns from OMI SP (black) and DOMINO (light blue) products. The seasonal cycle of the NO₂ surface concentrations measured in Kumpula air quality station are also shown in red. Note that VCDs and surface concentrations are reported on the left (molec. cm⁻² ) and right (ppb) y-axis, respectively.

The number of coincidences between OMI and the closest surface concentration measurement within 30 minutes are shown for each month on the top of both panels for both SP and DOMINO. The number of coincidences for the subset of Pandora observations are reported at the bottom of the left panel. The ground-based observations are sampled according to SP and DOMINO NO₂ products (continuous and dashed line, respectively). The error bars are estimated from the standard deviation of the mean. Only collocated observations with OMI CF<0.5 are taken into account.

The difference between OMI and Pandora NO₂ total columns are typically smaller than the uncertainties in satellite data. From the subset of collocated data in 2012, the uncertainty values in Pandora total columns are on average $3 \times 10^{14}$ molec. cm⁻² (or about 2%), while the total column median of the uncertainties is about one order of magnitude larger for OMI retrievals (15–30%).

Winter-autumn overpasses are often affected by clouds and also correspond to large SZA, increasing the uncertainty in the retrieval of the NO₂ total column. Data corresponding to spring-summer clear-sky days (SZA<65) show slightly smaller average difference (e.g., about 3% for SP) compared to the value obtained from the whole dataset. One would also expect better agreement for small pixels and short distance between Helsinki city center and the center of the satellite pixel. This is not clearly visible from Fig. 3 (bottom two panels). However, there are a few cases with very large difference (outliers in Fig. 3) between OMI and Pandora, which correspond to high values of distance and pixel area. For example, removing the overpasses with distance larger than 20 km does not reduce the difference between OMI and Pandora values. The largest differences between OMI and Pandora seem to be related to the effect of clouds and large SZAs more than to the pixel properties. It must





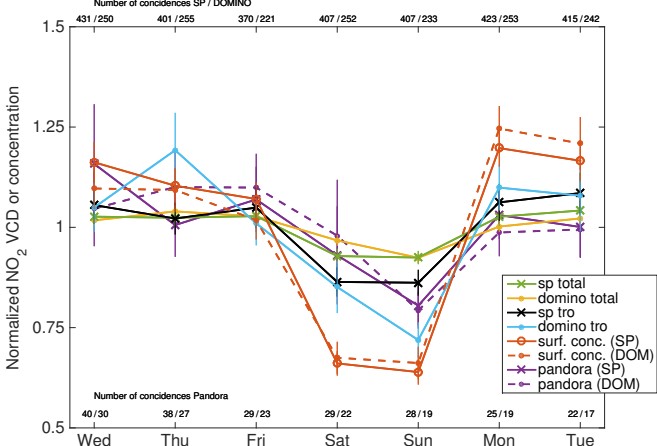

**Figure 5.** NO$_2$ weekly cycle from total and tropospheric columns from OMI SP (green and black, respectively) and DOMINO (yellow and light blue, respectively) products during 2006–2014. The weekly cycle of the NO$_2$ surface concentrations measured in Kumpula air quality station are also shown (red). The weekly cycle from collocated Pandora NO$_2$ total columns measured in Helsinki during 2012 are also shown (purple). The values for each day of the week are normalised with the weekly mean value in order to enhance the relative differences. The number of coincidences of OMI and the closest surface concentration measurement within 30 minutes are shown on the top of the figure for both SP and DOMINO. The number of coincidences for the subset of Pandora observations are reported at the bottom. The ground-based observations are sampled according to SP and DOMINO NO$_2$ products (continuous and dashed line, respectively). The error bars are estimated from the standard deviation of the mean. Only observations with OMI CF<0.5 are taken into account.

be also noted that the OMI pixels included as overpasses are distributed along the coastal line in the vicinity of Helsinki and might include the contribution of marine atmosphere (e.g., ship emissions).

## 3.2 Analysis of the seasonal and weekly cycle

Figure 4 (left panel) shows the monthly means of the NO$_2$ total columns from OMI SP and DOMINO overpasses in Helsinki
5 under almost clear-sky conditions (CF<0.5). The monthly means from Pandora total columns available in 2012 are shown for comparison. Figure 4 (right panel) includes the NO$_2$ tropospheric columns and the surface concentrations from Helsinki-Kumpula air quality station (located a few meters from the Pandora spectrometer). Only coincident OMI overpasses and surface concentration data are included in the calculation of the monthly means. Because Pandora data are available for one year, the number of coincidences for the Pandora observations is smaller than for OMI and concentration data (see inset numbers in
10 top and bottom axes in Fig. 4). Also, the number of coincidences for SP is different than for DOMINO because of different assumptions for snow covered surfaces and high solar zenith angles, which are recurring conditions at relatively high latitudes as in Helsinki (about 60°N). The error bars are determined as the standard deviation of the mean and thus are larger for decreasing number of coincidences.



The monthly means of tropospheric $NO_2$ and surface concentrations (Fig. 4 - right panel) show generally larger values in winter than in summer, as expected because of larger $NO_x$ emissions, shallower planetary boundary layer and longer lifetime in winter. However, the total column monthly means derived from OMI and Pandora total columns (Fig. 4 - left panel) do not clearly show such seasonal cycle. OMI DOMINO $NO_2$ total columns show different month-to-month variability compared

to SP, with SP monthly means generally closer to Pandora values and larger than DOMINO. Also, Pandora monthly means (purple lines in the left panel in Fig. 4) are characterized by larger error bars and variability than the other datasets, as result of the smaller number of data included in the calculation. The results are stongly affected by the fact that the number of available data is up to 2-3 times smaller in winter than in summer (mostly because of cloud screening, high SZAs and snow conditions). Thus, the monthly means calculated for winter months could be less representative of the actual $NO_2$ levels. In particular,

the DOMINO $NO_2$ monthly means for November and January include only the last and first half of the month, respectively, because of the screening of the scenes with high SZA values (larger than 80°). Also, for July and September, when relatively low monthly means are obtained from Pandora observations, the number of coincidences is about three times smaller than the other summer months, suggesting that these values are less statistically reliable. It must be also pointed out that sampling Pandora and surface concentration observations according to SP or DOMINO datasets (continuous and dashed purple and

orange lines in Fig. 4, respectively) does not substantially change the monthly means in summer (differences smaller than the error bars), while larger differences can be noticed for winter months.

Figure 5 shows the weekly cycle of the $NO_2$ total and tropospheric columns from OMI SP and DOMINO datasets. Also the weekly cycle from Pandora $NO_2$ total columns and surface concentrations from Helsinki-Kumpula air quality station are included for comparison. The values are normalised with the weekly mean value in order to enhance the relative differences.

The data correspond to the same overpasses presented in Fig. 4. All datasets show smaller values in the weekend compared to the other weekdays. This is expected because of the reduced emissions from car traffic and industrial activity during the weekend. $NO_2$ levels are usually slightly lower on Sunday than on Saturday. The amplitude of the weekly cycle can be quantified as the percentage reduction between weekend and weekdays. The $NO_2$ surface concentration is on average 40% smaller on the weekend than on the weekdays. The amplitude of the weekly cycle become increasingly smaller for tropospheric

and total columns (15-30% and 7-9%, respectively, from OMI and 24% for Pandora total columns). This dampening in the weekly cycle is expected because the surface concentrations are closer to the actual emission changes. The tropospheric column weekly cycle in Helsinki is similar but slightly smaller than the Europe average (amplitude about 40%) as derived by Beirle et al. (2003). The weekly cycle values slightly increase when also cloudy scenes are taken into account, probably because of the larger amount of winter observations included in the calculation.

**4  Summary and conclusions**

In this work, OMI $NO_2$ products have been compared against ground-based observations in Helsinki in order to evaluate their applicability for air quality monitoring at high latitudes. The main results of this comparison are summarised below:





- OMI SP $NO_2$ total columns agree on average within about $\pm 5\%$ under clear sky conditions with ground-based observations obtained from Pandora spectrometer. The largest differences are observed for autumn-winter days, which are characterised by cloudy conditions and large SZAs. OMI DOMINO $NO_2$ data show slightly smaller absolute values of $NO_2$ total column than SP, mainly because of different air mass factor values.

- OMI $NO_2$ total and tropospheric columns show a similar weekly cycle as the $NO_2$ surface concentrations in Helsinki, with smaller values in the weekend compared to the weekdays. Also, the weekly cycle observed from OMI total columns compare well with the one obtained from Pandora measurements.

- OMI tropospheric $NO_2$ seasonal cycle is similar to the one obtained from surface concentrations, while the total columns are strongly affected by the scarse amount of data included in the monthly mean calculation. During autumn-winter most
of the data are screened as cloudy and the resulting monthly means are typically characterised by large error bars.

- OMI cloud fraction values are used for selecting almost clear-sky scenes. OMI CFs give the same cloud-screening results as ground-based cloud cover information in more than 80% of the cases.

In summary, despite relatively low $NO_2$ levels in Helsinki and frequent cloudy conditions, OMI $NO_2$ data have been able to realistically represent air quality features at the surface also at such high latitudes site. The average differences are comparable
to those obtained for middle latitude sites (see, e.g. Table 2 in Lamsal et al., 2014), with Pandora total columns usually higher than OMI retrievals.

The main limitations in using satellite data at high latitudes are related to the reduced light-hours and large amount of cloudy pixels during autumn-winter season. The weekly and seasonal cycles reported in this work are obtained mostly for spring or summertime conditions. Smaller pixel size would reduce the amount of scenes screened as cloudy. A much smaller footprint
will be achieved by the upcoming TROPOMI instrument (launch planned on October 2016), which will provide $NO_2$ observations with improved spatial resolution (7x7 $km^2$ at nadir) and signal-to-noise ratio. These features will be particularly important for monitoring air quality of relatively small sources such as the city of Helsinki and will increase the amount of cloud-free pixels available for future analysis. Further studies will aim at validating TROPOMI observations when available using measurements from a new Pandora instrument recently installed at FMI. Also, the effect of the snow/ice surface reflectivity
information on the retrieval will be analysed.

*Acknowledgements.* This work of I.I. was founded by ILMA-project (Applications of $NO_2$ satellite observations at high latitudes for monitoring air quality) within the ESA Living Planet Programme. J.T. was partially funded by Academy od Finalnd project INQUIRE. The authors acknowledge the NASA Earth Science Division and KNMI for funding the OMI $NO_2$ development and archiving of standard and DOMINO products, respectively. The authors also thank the Atmospheric Science Department of the University of Helsinki for providing
surface concentration measurements through the SmartSMEAR download tool.





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
