# Peer review of "Comparison of OMI NO2 observations and their seasonal and weekly cycles with ground-based measurements in Helsinki"

_Atmospheric Measurement Techniques, 2016_

## Referee Comment (RC1) · Anonymous Referee #1 · 19 Jul 2016

The paper presents the comparison of NO2 column measured by Pandora sun-photometer with two OMI satellite products. It is a well presented paper that demonstrates the capabilities of Pandora for satellite NO2 validation. The subject is relevant for AMT and the quality of the paper is good to be accepted for publication in AMT.

However, before publication I recommend to the authors to address the following issues:

General comments:

1. Pandora was installed in 2012, but only one year of data was discussed. If more data are available, I suggest the authors use all of them.

[Figure]

2. Pandora is a relatively new instrument. It would be useful to include some technical details in the paper. For example, what spectral window was used for the spectral fitting? Were any attenuation filters used? If so, are the results for different filters consistent? A temperature correction is mentioned on p. 4. How it was done? Is it a part of Pandora's operational software of something developed by the authors.

3. The authors do not mention any diurnal NO2 total column variations. It would be useful to have some information about them for satellite data interpretation.

Specific comments:

1, 2: Change to "NASA standard product (SP) and KNMI DOMINO product"

3, 18: What OMI data product is discussed here? SP?

4, 6: "Temperature correction" What temperature data are used for this correction? Climatological? How large is the correction?

5, 3: Figure 2. There are too many dots and symbols on this plot. Perhaps it is better not to show bad data.

6, 1: Figure 3. You could add one more panel that shows the difference vs. time using the colorscale that represent NO2 values themselves

6, 6: The supplementary material contains only one figure. Add S1 to Figure 3 and drop the supplement.

7, : Figure 4. It is very difficult to see overlapping error bars. Shift them slightly or use different thickness for the error bars.

7, 1: This sentence is confusing. The difference between individual OMI and Pandora measurements cannot be smaller than the uncertainties of individual OMI measurements. Or, you are talking about systematic differences here?

7, 9: Figure 3 is not enough for such statement. Could you calculate the standard

deviation of the OMI-Pandora difference for small and large OMI pixels?

8, Figure 5. Do you really need SP total and DOMINO total in this figure? Also, you could drop surf.con. (DOM) and Pandora (DOM) since they are very similar to the surf.con (SP) and Pandora (SP)

10, 12: Be more specific here: "OMI CFs below 0.5 valued give the same cloud-screening results as the ground-based cloud cover below 5/8 condition in more than 80% of the cases."

---

## Referee Comment (RC2) · Anonymous Referee #2 · 1 Aug 2016

"Comparison of OMI NO2 observations and their seasonal and weekly cycles with ground-based measurements in Helsinki" by Ialongo et al. (2016) presents a comparison of OMI NO2 with ground-based measurements over Helsinki. This is a nicely conceived study and, as the authors point out, it is one of a handful that focuses on higher latitudes. If the authors consider the issue I elaborate on below and take steps to address it in a meaningful way then I recommend publication.

Main issue:

This study finds agreement to within about 5% between OMI-SP and Pandora total NO2 VCDs. While remarkable, I believe this is not real and is likely the result of cancelling errors. It is now well established that OMI SCDs are high biased by something like 15-

30% (higher % corresponds to smaller SCD) (see Belmonte-Rivas, 2014; Marchenko et al., 2015; van Geffen et al., 2015), and that this bias largely gets incorporated into stratospheric VCDs. Indeed, this has prompted a large effort to redo the spectral fitting by both NASA and KNMI (Marchenko et al., 2015; van Geffen et al., 2015). By comparing SP (and DOMINO) with ground-based using total columns, the OMI total VCD is high biased by something like 25%.

This is likely being cancelled by a spatial resolution effect. Your Pandora is located in the middle of a relative small (on the order of 30 km) NO2 hotspot, at least in an average sense. Even on a fine grid, the effect of the coarse OMI pixel resolution will be to always include small, upwind columns. Thus one would expect the Pandora, with an effective spatial resolution on the order of a couple km, to be systematically higher. Likely exacerbating this, it appears you use even the largest track positions which have cross-track resolutions of $\sim$150 km. You can see a hint of this in your Figure 3c, but the effect will be present since even the smallest pixels are 24 km across. This effect is why the slope of an OMI vs. ground-based scatterplot is typically 0.4-0.7 (see, McLinden et al., 2014 or Kharol et al., 2015, but there are many other examples). There is no reason to expect a slope of unity here given the relative location of NOx sources, Pandora, and OMI pixel sizes. I tried to demonstrate this in Figure 1, below, using your average map and proxies for OMI pixels which clearly are sampling outside the peak NO2 area.

Knowing that there is a large systematic error in the stratospheric VCDs, it is not reasonable to keep this comparison as-is. Here are some possible ways to remedy this:

1. Wait until the next version of the SP product is available which corrects the spectral fitting issue. My understanding its release is imminent, and since you have SP co-authors, they might give you early access.

2. Compare tropospheric VCDs. This means you will have to estimate the stratospheric portion of the Pandora VCD and remove it. You could use another satellite, SCIAMACHY, OSIRIS, or other, or use a model.

3. Scale the OMI stratospheric VCD by ∼0.75 or 0.8 and recalculate total VCD. You might be able to find a better scaling (but 0.75-0.8 should be about right) using papers such as Marchenko et al., 2015 and Adams et al., 2016.

4. One of your co-authors is Nick Krotkov. He might have another suggestion.

The spatial resolution issue is harder to address, although it needs to be mentioned and cited as a source of comparison bias. One could estimate the effect by getting a map of gridded emissions, say from HTAP, and smoothing it to OMI resolution (∼50 x 30), and then comparing the smoothed and unsmoothed VCDs at the location of the Pandora.

Other points:

1. Show a scatterplot of OMI vs. Pandora.

2. What precisely did you do to filter the OMI data? E.g., were snow covered pixels removed? You hint at this in the analysis, but please state.

3. Add lat/lon and a scale to figure 1. The OMI pixel outlines would also be instructive (i.e., a nicer version of what I did above).

References:

Adams, C., E. N. Normand, C. A. McLinden, A. E. Bourassa, D. A. Degenstein, N. A. Krotkov, M. Belmonte Rivas, K. F. Boersma, and H. Eskes, Limb–nadir matching using non-coincident NO2 observations: Proof of concept and the OMI-minus-OSIRIS prototype product, Atmos. Meas. Tech., doi:10.5194/amt-2016-138, 2016.

Belmonte Rivas, M., Veefkind, P., Boersma, F., Levelt, P., Eskes, H. and Gille, J.: Intercomparison of daytime stratospheric NO2 satellite retrievals and model simulations, Atmos. Meas. Tech., 7, 2203–2225, doi:10.5194/amt-7-2203-2014, 2014.

Kharol, S. K., R. V. Martin, S. Philip, B. Boys, L. N. Lamsal, M. Jerrett, M. Brauer, D. L. Crouse, C. McLinden, R.T. Burnett, Assessment of the magnitude and recent trends in

satellite-derived ground-level nitrogen dioxide over North America, Atmos. Env., 118, 236-245, 2015.

Marchenko, S., Krotkov, N. A., Lamsal, L. N., Celarier, E. A., Swartz, W. H. and Bucsela, E. J.: Revising the slant column density retrieval of nitrogen dioxide observed by the Ozone Monitoring Instrument, J. Geophys. Res. Atmos., 120(11), 1–23, doi:10.1002/2014JD022913, 2015.

McLinden, C. A., V. Fioletov, K. F. Boersma, S. Kharol, N. Krotkov, P. A. Makar, R. V. Martin, J. P. Veefkind, and K. Yang, Satellite retrievals of NO2 and SO2 over the Canadian oil sands and comparisons with surface measurements, Atmos. Chem. Phys., 14, 3637–3656, doi:10.5194/acp-14-3637-2014, 2014.

van Geffen, J. H. G. M., Boersma, K. F., Van Roozendael, M., Hendrick, F., Mahieu, E., De Smedt, I., Sneep, M. and Veefkind, J. P.: Improved spectral fitting of nitrogen dioxide from OMI in the 405–465 nm window, Atmos. Meas. Tech., 8, 1685–1699, doi:10.5194/amt-8-1685-2015, 2015.

**May-Sep 2005-2014**

Figure showing NO2 column density map over Finland/Estonia region with cities Turku, Helsinki, Kouvola, Tallinn, Narva labeled. Three representative OMI pixels overlaid as red, blue, and black rectangles near Helsinki. An 80 km distance marked between Helsinki and Tallinn.

NO$_2$ column density (molec./cm$^2$)  ×10$^{15}$

**Fig. 1.** Figure 1 from Ialongo et al., overlaid with three representative OMI pixels: red/small (13 km x 27 = 350 km2), medium/blue (15 km x 38 km = 570 km2), and large/black (20 km x 78 km = 1560 km2).

---

## Author Comment (AC1) · 15 Sep 2016

Answer to Anonymous Referee #1 for the manuscript "Comparison of OMI NO2 observations and their seasonal and weekly cycles with ground-based measurements in Helsinki" by Ialongo et al. (2016)

The authors thank the Referee #1 for the useful comments. The point-to-point answer is provided as follows. The Referee comments are in *Italics*, while the answer by the authors is in **Bold Roman**. Please find the author's changes in the track-changed revised manuscript.

*1) Pandora was installed in 2012, but only one year of data was discussed. If more data are available, I suggest the authors use all of them.*

**Pandora #21 was operational in Helsinki from late 2011 to early 2013. We use the data from the full year 2012, discarding the scattered measurements from other years. Currently, a different Pandora instrument is working at the same place at FMI but the field calibration procedures (requiring measurements during several sunny days) have not yet been completed. We aim at using this new set of data in the near future for validating NO2 retrievals from both OMI and TROPOMI instruments.**

*2) Pandora is a relatively new instrument. It would be useful to include some technical details in the paper. For example, what spectral window was used for the spectral fitting? Were any attenuation filters used? If so, are the results for different filters consistent? A temperature correction is mentioned on p. 4. How it was done? Is it a part of Pandora's operational software of something developed by the authors.*

**Pandora NO2 retrievals are based on the operational software as described by Herman et al. (2009). This is why we refer to that paper for the technical details. We updated PANDORA discussion on page 4 including more details as follows:**

**" The algorithm first derives the relative NO2 slant columns densities (SCDs) by DOAS spectral fitting technique (e.g., Cede et al., 2006) in 370-500nm (see Fig. 5 in Herman et al., 2009) and converts them to absolute SCDs using statistically estimated reference spectrum obtained from on-site PANDORA measurements by Langley-extrapolation technique. Pandora SCD retrieval employs a temperature correction to the cross-sections used in the spectral fitting procedure as described in Eq.1 by Herman et al. (2009), based on modelled monthly average $NO_2$ and temperature profiles and high-resolution temperature-dependent cross sections by Vandaele et al. (1988), as also for OMI NO2 retrievals."**

**Cede, A., J. Herman, A. Richter, N. Krotkov, and J. Burrows (2006), Measurements of nitrogen dioxide total column amounts at Goddard Space Flight Center using a Brewer spectrometer in direct sun mode, J. Geophys. Res., 111, D05304, doi:10.1029/2005JD006585.**

**Vandaele, A. C., C. Hermans, P. C. Simon, M. Carleer, R. Colin, S. Fally, M. F. Mérienne, A. Jenouvrier, *and* B. Coquart (1998), Measurements of the NO₂ absorption cross-section from 42000 cm⁻¹ to 10000 cm⁻¹ (238–1000 nm) at 220 K and 294 K, J. Quant. Spectrosc. Radiat. Transfer, 59(3–5), 171–184.**

*3) The authors do not mention any diurnal NO2 total column variations. It would be useful to have some information about them for satellite data interpretation.*

**We agree with reviewer's comment with the following caveat. The diurnal dependences of NO2 in PBL and stratosphere are different, driven by un-related processes. Since PANDORA measures total column (stratospheric plus tropospheric) the interpretation of the measured diurnal dependence may not be straightforward. The following text has been added in section 3.1:**

**"Figure 2 also illustrates the measured diurnal variations in NO2 total columns. The daily cycle is highly variable from day-to-day, depending on several factors, such as the diurnal cycle of anthropogenic NOx emissions, NOx photochemistry, relative contribution from stratospheric columns, as well as changing meteorological conditions. Under clear sky conditions, Pandora NO2 total columns show peaks in the morning or in the afternoon (as would be expected from increased car traffic during the rush hours and small contribution from stratospheric columns). Sometimes, very low NO2 total columns are observed throughout the day, as for example on 1 May 2012 (first panel in Fig. 2), probably because of the wind patterns. OMI overpasses occur between 12 and 15:30 local time (outside the rush hours), when relatively low tropospheric NO2 levels are expected."**

*Specific comments:*
*1, 2: Change to "NASA standard product (SP) and KNMI DOMINO product"*
**Corrected**

*3, 18: What OMI data product is discussed here? SP?*
**Yes, corrected.**

*4, 6: "Temperature correction" What temperature data are used for this correction? Climatological? How large is the correction?*
**See point 2)**

*5, 3: Figure 2. There are too many dots and symbols on this plot. Perhaps it is better not to show bad data.*
**Corrected**

*6, 1: Figure 3. You could add one more panel that shows the difference vs. time using the colorscale that represent NO2 values themselves*

**Corrected. We also changed the text accordingly.**

*6, 6: The supplementary material contains only one figure. Add S1 to Figure 3 and drop the supplement.*

**There is now a second figure in the supplement. We prefer to keep the supplement to avoid redundant information in the main text.**

*7, : Figure 4. It is very difficult to see overlapping error bars. Shift them slightly or use different thickness for the error bars.*

**In Fig.5 of the revised manuscript, we removed the surf. con. (DOM) and Pandora (DOM) lines, to be consistent with figure 5 and reduce redundant information. This should also help visualizing the error bars.**

7, 1: This sentence is confusing. The difference between individual OMI and Pandora measurements cannot be smaller than the uncertainties of individual OMI measurements. Or, you are talking about systematic differences here?

**We removed this phrase while also changed the text to answer to comments from referee #2. We discuss the uncertainty more accurately in the revised manuscript.**

7, 9: Figure 3 is not enough for such statement. Could you calculate the standard deviation of the OMI-Pandora difference for small and large OMI pixels?

**There is indeed no large improvement in the mean rel. dif. (about 1 percent closer to zero), when removing pixels at the edge of the swath but we further illustrate the effect of the pixel size in Fig. S2 (as requested also by Referee #2) in the supplement of the revised manuscript. We added this sentence in section 3.1:**
**"For example, the average relative difference between OMI SP and Pandora derived using relatively small pixels (cross-track position 6-55) is (-5±25)%, about one percentage point closer to zero than for the whole data set (-6±25)%."**

8, Figure 5. Do you really need SP total and DOMINO total in this figure? Also, you could drop surf.con. (DOM) and Pandora (DOM) since they are very similar to the surf.con (SP) and Pandora (SP)

**In Fig.6 of the revised manuscript, we drop the surf.con. (DOM) and Pandora (DOM) lines since they are redundant but we prefer to keep the total column to allow the comparison with Pandora total columns.**

10, 12: Be more specific here: "OMI CFs below 0.5 valued give the same cloud-screening results as the ground-based cloud cover below 5/8 condition in more than 80% of the cases."

**Corrected**

---

## Author Comment (AC2) · 15 Sep 2016

Answer to Anonymous Referee #2 for the manuscript "Comparison of OMI NO2 observations and their seasonal and weekly cycles with ground-based measurements in Helsinki" by Ialongo et al. (2016)

The authors thank the Referee #2 for the constructive comments that contributed to improve the manuscript. The point-to-point answer is provided as follows. The Referee comments are in *Italics*, while the answer by the authors is in **Bold Roman**. Please find the author's changes in the track-changed revised manuscript.

*1) Main issue:*
*This study finds agreement to within about 5% between OMI-SP and Pandora total NO2 VCDs. While remarkable, I believe this is not real and is likely the result of cancelling errors. It is now well established that OMI SCDs are high biased by something like 15-30% (higher % corresponds to smaller SCD) (see Belmonte-Rivas, 2014; Marchenko et al., 2015; van Geffen et al., 2015), and that this bias largely gets incorporated into stratospheric VCDs. Indeed, this has prompted a large effort to redo the spectral fitting by both NASA and KNMI (Marchenko et al., 2015; van Geffen et al., 2015). By comparing SP (and DOMINO) with ground-based using total columns, the OMI total VCD is high biased by something like 25%.*

**This aspect in now discussed in the text (see at the end of this file). In order to evaluate the effect of the new spectral fitting, both the version 2.1 and the upcoming version 3 (preliminary version) of OMI NO2 columns are now compared with Pandora observations in the scatterplot in Fig.4 of the revised manuscript. Similar updates are not yet available from DOMINO product; thus, we use version 2 for the rest the analysis. See also point 3).**

*2) This is likely being cancelled by a spatial resolution effect. Your Pandora is located in the middle of a relative small (on the order of 30 km) NO2 hotspot, at least in an average sense. Even on a fine grid, the effect of the coarse OMI pixel resolution will be to always include small, upwind columns. Thus one would expect the Pandora, with an effective spatial resolution on the order of a couple km, to be systematically higher. Likely exacerbating this, it appears you use even the largest track positions, which have cross-track resolutions of 150 km. You can see a hint of this in your Figure 3c, but the effect will be present since even the smallest pixels are 24 km across. This effect is why the slope of an OMI vs. ground-based scatterplot is typically 0.4-0.7 (see, McLinden et al., 2014 or Kharol et al., 2015, but there are many other examples). There is no reason to expect a slope of unity here given the relative location of NOx sources, Pandora, and OMI pixel sizes. I tried to demonstrate this in Figure 1, below, using your average map and proxies for OMI pixels, which clearly are sampling outside the peak NO2 area.*

**The effect of the relatively coarse OMI spatial resolution is now discussed in the text of the revised manuscript and the distribution of the NOx emission over Helsinki area is reported in figure S2 in the supplementary material. See also point 4). The outlines of OMI pixels with different size are overlapped in order to illustrate the dilution effect due to the relatively coarse OMI spatial resolution. Averaging over OMI pixel area can reduce**

the observed signal from the NOx emission by about 20 to 80%, depending on the size of the pixel.
Also, Vasilikov et al. (2016) reported that correcting for the effect of the different observing geometries on the surface reflectivity might produce a sizeable change in the OMI vertical column, which might in turn affect the observed bias between OMI and Pandora.

These effects are discussed in the revised manuscript (see at the end of this file).

Vasilkov, A., Qin, W., Krotkov, N., Lamsal, L., Spurr, R., Haffner, D., Joiner, J., Yang, E.-S., and Marchenko, S.: Accounting for the effects of surface BRDF on satellite cloud and trace-gas retrievals: A new approach based on geometry-dependent Lambertian-equivalent reflectivity applied to OMI algorithms, Atmos. Meas. Tech. Discuss., doi:10.5194/amt-2016-133, in review, 2016.

*3) Knowing that there is a large systematic error in the stratospheric VCDs, it is not reasonable to keep this comparison as-is. Here are some possible ways to remedy this:*
*1. Wait until the next version of the SP product is available which corrects the spectral fitting issue. My understanding its release is imminent, and since you have SP co-authors, they might give you early access.*
*2. Compare tropospheric VCDs. This means you will have to estimate the stratospheric portion of the Pandora VCD and remove it. You could use another satellite, SCIAMACHY, OSIRIS, or other, or use a model.*
*3. Scale the OMI stratospheric VCD by 0.75 or 0.8 and recalculate total VCD. You might be able to find a better scaling (but 0.75-0.8 should be about right) using papers such as Marchenko et al., 2015 and Adams et al., 2016.*
*4. One of your co-authors is Nick Krotkov. He might have another suggestion.*

We now compare the new version 3 (preliminary data) of the SP product together with the current version 2.1, in order to evaluate the effect of the improved spectral fitting. Fig. 4 in the revised manuscript shows the results of the comparison as scatterplot. The results show median relative difference between OMI SP V3 and Pandora of about -32% (instead of -5% for Version 2.1). The slopes are 0.39 and 0.49 for V2.1 and V3, respectively and the correlation moderate (r=0.51 for both versions). This result is as expected and consistent with the literature mentioned in point 1. Future studies will focus on more comprehensive validation of V3, when such data will be available in their final form from both SP and DOMINO algorithms.

*4) The spatial resolution issue is harder to address, although it needs to be mentioned and cited as a source of comparison bias. One could estimate the effect by getting a map of gridded emissions, say from HTAP, and smoothing it to OMI resolution (50 x 30), and then comparing the smoothed and unsmoothed VCDs at the location of the Pandora.*

**As suggested we now include a map of the emission in Helsinki area (Fig S2 in the supplement) in order to evaluate the effect of the relatively coarse OMI resolution. We also plot three examples of OMI pixels on top of the map in order to illustrate the dilution effect due to the coarse pixels. Also we report now the results of the median difference only including relatively small pixels (from 6 to 55). See the text at the end of this file.**

*Other points:*
*5). Show a scatterplot of OMI vs. Pandora.*
**This in now shown in Figure 4 of the revised manuscript, including also the retrievals from SP version 3. See also point 3).**

*6). What precisely did you do to filter the OMI data? E.g., were snow covered pixels removed? You hint at this in the analysis, but please state.*
**In Fig. 3 there is no strong filtering applied (only surface reflectance below 0.2 and distance below 50 km), because we want to illustrate the effect of the different parameters on the comparison. When calculating the median differences we screen also for cloudy conditions. We report also the median relative difference separately for small SZA (<65deg) and small pixels (cross-track positions 6-55). We now mention this explicitly in the text.**

*7). Add lat/lon and a scale to figure 1. The OMI pixel outlines would also be instructive (i.e., a nicer version of what I did above).*
**We added lat/lon and scale. We added the pixel outlines on the Fig. S2 in the revised manuscript, together with the NOx emission map, in order to illustrate the effect of different sized pixels. So, we omitted those from Fig. 1 to avoid redundancy.**

*References:*
*Adams, C., E. N. Normand, C. A. McLinden, A. E. Bourassa, D. A. Degenstein, N. A. Krotkov, M. Belmonte Rivas, K. F. Boersma, and H. Eskes, Limb–nadir matching using non-coincident NO2 observations: Proof of concept and the OMI-minus-OSIRIS prototype product, Atmos. Meas. Tech., doi:10.5194/amt-2016-138, 2016.*
*Belmonte Rivas, M., Veefkind, P., Boersma, F., Levelt, P., Eskes, H. and Gille, J.: Intercomparison of daytime stratospheric NO2 satellite retrievals and model simulations, Atmos. Meas. Tech., 7, 2203–2225, doi:10.5194/amt-7-2203-2014, 2014.*
*Kharol, S. K., R. V. Martin, S. Philip, B. Boys, L. N. Lamsal, M. Jerrett, M. Brauer, D. L. Crouse, C. McLinden, R.T. Burnett, Assessment of the magnitude and recent trends in satellite-derived ground-level nitrogen dioxide over North America, Atmos. Env., 118, 236-245, 2015.*
*Marchenko, S., Krotkov, N. A., Lamsal, L. N., Celarier, E. A., Swartz, W. H. and Bucsela, E. J.: Revising the slant column density retrieval of nitrogen dioxide observed by the Ozone Monitoring Instrument, J. Geophys. Res. Atmos., 120(11), 1–23, doi:10.1002/2014JD022913, 2015.*
*McLinden, C. A., V. Fioletov, K. F. Boersma, S. Kharol, N. Krotkov, P. A. Makar, R. V.*

*Martin, J. P. Veefkind, and K. Yang, Satellite retrievals of NO2 and SO2 over the Canadian oil sands and comparisons with surface measurements, Atmos. Chem. Phys., 14, 3637–3656, doi:10.5194/acp-14-3637-2014, 2014.*
*van Geffen, J. H. G. M., Boersma, K. F., Van Roozendael, M., Hendrick, F., Mahieu, E., De Smedt, I., Sneep, M. and Veefkind, J. P.: Improved spectral fitting of nitrogen dioxide from OMI in the 405–465 nm window, Atmos. Meas. Tech., 8, 1685–1699, doi:10.5194/amt-8-1685-2015, 2015.*

**The references are added in the text**

**The following discussion about the sources of errors affecting the comparison will be added to the revised manuscript (section 3.1):**

**"For example, the average relative difference between OMI SP and Pandora derived using relatively small pixels (cross-track position 6-55) is (-5±25)%, about one percent closer to zero than for the whole data set (-6±25)%.**
**These average values are the result of different effects, potentially cancelling each other. For example, Belmonte Rivas et al. (2014), Marchenko et al. (2015) and van Geffen et al. (2015) reported that OMI slant column densities are high biased by about 10-40%, producing an overestimation in the stratospheric vertical columns of the same order of magnitude (Adams et al., 2016). This causes the OMI retrievals to overestimate the total columns when compared to Pandora measurements. Marchenko et al. (2015) and van Geffen et al. (2015) proposed revisions of the spectral fitting in the OMI NO2 retrieval algorithm, which reduce the slant column densities by 10-35%, bringing them closer to independent measurements. The next-generation the OMI NO2 product (Version 3) accounts for this improved spectral fitting. Thus, in order to evaluate this positive bias, we compare Pandora total columns to a subset of data including both SP V2.1 and V3. The results are presented in the scatter plot in Fig. 4 for cross-track positions 6-55. The median relative difference for V3 is (-32±18)% and it is much larger than for V2.1 (-5%). The linear fit slopes are 0.49 and 0.39 for V2.1 and V3, respectively, and the correlation is moderate (r = 0.51 for both datasets). Such values are comparable to the values obtained for example by McLinden et al. (2014) and Kharol et al. (2015) using in-situ surface observations. Slope values close to the unity are not expected, because of the different spatial resolution of satellite- and ground-based observations.**
**The difference between the OMI pixel and the relatively smaller Pandora FOV is indeed expected to cause an underestimation of the total column by OMI. This effect is analysed in Fig. S2 in the supplement, where the 2010 EDGARv4.3.1 NOx emission map (available at http://edgar.jrc.ec.europa.eu) over Helsinki is presented. The outlines of three OMI pixels with different size (cross-track position 17, 53 and 60 with areas of 430, 870 and 3560 km2, respectively) are overlapped to the emission map, in order to illustrate the effect of the relatively coarse spatial resolution. One must note that, because of the row anomaly, the smallest pixels at the center of the swath (close to cross-track position 30)**

are not 20 available for the comparison. The emission estimate at the location of Pandora (red dot in Fig. S2) is about 5 ktons yr−1. When averaging over OMI pixel area, the emission values decrease while the size of the pixel increases. The emissions are about 20, 40 and 80% smaller than the value at the Pandora location for pixel 17, 53 and 60, respectively. This difference in emissions is at least partially transferred to the vertical column (by a factor of about 0.8 according to Lamsal et al. (2011)). Similarly, Irie et al. (2012) and Lin et al. (2014) found large discrepancies between space- and ground-based measurements especially over areas with high $NO_2$ spatial inhomogeneity, due to their different spatial representativeness. The comparison can be also affected by the position of the center of the OMI pixel compared to the ground-based station, because different pixels sample different areas around the ground-based station. The OMI pixels included as overpasses are distributed along the coastal line in the vicinity of Helsinki and might include the contribution of marine atmosphere (e.g., ship emissions) or other pollution sources over land.

Also, occasionally, Pandora $NO_2$ values build up to relatively high pollution levels (over $1.5×10^{16}$ molec.cm−2). This likely occurs when the ground-based station is downwind from a main high traffic street. The difference between OMI 30 and Pandora total columns shows relatively large negative values (OMI smaller than Pandora) for relatively large Pandora total columns (Fig. 3 - bottom panel), hinting that OMI is less able to reproduce such episodes of localised and elevated pollution because of the coarse pixel size. Overall, Pandora $NO_2$ total columns are expected to be larger than OMI retrievals because of the effect of the coarse OMI spatial resolution. This might partly cancel the positive bias caused by the overestimation of the stratospheric columns.

Furthermore, Vasilkov et al. (2016) analysed the effect of the varying observation geometry on the $NO_2$ vertical column retrieval. They found that, replacing the current OMI-based Lambertian-equivalent reflectivity (LER) climatology (Kleipool et al., 2008) used in OMI $NO_2$ algorithms with a high-resolution geometry-dependent LER based on MODIS (Moderate Resolution Imaging Spectroradiometer) observations, causes an overall increase in the vertical column values over a test study orbit in America. This effect could further change the bias we observe between OMI and Pandora retrievals."

Lin, J.-T., Martin, R. V., Boersma, K. F., Sneep, M., Stammes, P., Spurr, R., Wang, P., Van Roozendael, M., Clemer, K., and Irie, H.: Retrieving tropospheric nitrogen dioxide from the Ozone Monitoring Instrument: effects of aerosols, surface reflectance anisotropy, and vertical profile of nitrogen dioxide, Atmos. Chem. Phys., 14, 1441-1461, doi:10.5194/acp-14-1441-2014, 2014.

Irie, H., Boersma, K. F., Kanaya, Y., Takashima, H., Pan, X., and Wang, Z. F.: Quantitative bias estimates for tropospheric $NO_2$ columns retrieved from SCIAMACHY, OMI, and GOME-2 using a common standard for East Asia, Atmos. Meas. Tech., 5, 2403-2411, doi:10.5194/amt-5-2403-2012, 2012.

**Kleipool, Q. L., M. R. Dobber, J. F. de Haan, and P. F. Levelt (2008), Earth surface reflectance climatology from 3 years of OMI data, J. Geophys. Res., 113, D18308, doi:10.1029/2008JD010290.**